# Fast Unsupervised Generative Design for Structural Topology Optimization

**Ethan Herron**    **Anushrut Jignasu**    **Jaydeep Rade**    **Xian Yeow Lee**
**Aditya Balu**    **Adarsh Krishnamurthy**    **Soumik Sarkar** †

Iowa State University
†soumiks@iastate.edu

## Abstract

Exploring the intersection of generative design and structural topology optimization has been a popular research area recently. Existing structural optimization methods have been shown to generate high-performance and aesthetically pleasing structures, but at a tremendous computational cost. The rapidly advancing field of deep learning, particularly generative modeling, has substantial potential to tackle the structural generative design problem. Previous works have utilized deep generative models for highly specific cases, ranging from a small set of loading conditions to heavily relying on supervised loss functions for training. We propose a new method targeted at generating near-optimal structures over a wide variety of initial conditions in a completely unsupervised manner. We accomplish this by implementing a novel Generative Adversarial Network (GAN) framework to generate densities that match our given target distribution and encode extremely efficient latent representations of the initial physical conditions of the sample. The target distribution used in this work comes from data generated via the solid isotropic material condition with penalization (SIMP) topology optimization algorithm. Our results show that the proposed framework can generate similar structures to those found using the SIMP optimization algorithm, which consequently demonstrates the potential variability in solution spaces for arbitrary problems in generative design.

## Introduction

Topology Optimization refers to optimizing the structure with regards to various constraints such as weight, structural capacity, etc., from an initial geometry along with physical boundary conditions. Currently, there are reliable and efficient, performance-wise, algorithms available and frequently used in practice (Bendsøe 1989).

Formally, topology optimization is represented as:

$$\begin{aligned}
\text{minimize: } & C(U) \\
\text{subject to: } & \mathbf{K}\mathbf{U} = \mathbf{F} \\
& g_i(\mathbf{U}) \leq 0.
\end{aligned} \quad (1)$$

Here, $C(U)$ refers to the objective function of topology optimization. In the case of structural topology optimization, this

is the compliance of the system,

$$C = \int_{\Omega \in \mathcal{S}} bu \, d\Omega + \int_{\tau \in d\mathcal{S}} tu \, d\tau \quad (2)$$

where $b$ represents the body forces, $u$ displacements, $t$ surface traction, and $\Omega$ and $\tau$ are volume and surface representations of solid. The constraint $g_i(U)$ includes a volume fraction constraint, $g_i = (v/v_0) - v_f$. Since this optimization is performed for every element in the mesh, the combinatorial optimization is computationally intractable. Naturally, an alternative solution is to represent the same set of equations above as a function of density $\rho$ for every element.

$$\begin{aligned}
\text{Minimize: } & C(\rho, U) \\
\text{subject to: } & \mathbf{K}(\rho)\mathbf{U} = \mathbf{F} \\
& g_i(\rho, \mathbf{U}) \leq 0 \\
& 0 < \rho \leq 1
\end{aligned} \quad (3)$$

This design problem is relaxed using solid isotropic material condition with penalization (often called SIMP (Bendsøe 1989)), where the stiffness for each element may be described as, $E = E_{min} + \rho^p(E_{max} - E_{min})$. Here, $p$ is the parameter used for penalizing the element density to be closer to $1.0$. Structural topology optimization methods, SIMP, in particular, are still subject to multiple computationally expensive finite element calls during the optimization process. Additionally, these optimization methods all produce comparable but different structures. This shows that there may be a larger solution space for optimal structures than one would assume.

The downside is that traditional topology optimization algorithms can be extremely computationally expensive. There has been significant interest in utilizing deep learning algorithms to achieve similar performance with drastically less computational costs. Using deep learning techniques for solving topology optimization has been pursued aggressively. Direct supervised approaches to predict the final optimized topology when given the initial conditions, i.e., the initial strain energy and the design constraint, referred to in this paper as initial volume fraction, have produced exceptional results (Abueidda, Koric, and Sobh 2020). Other approaches have taken a more iterative approach that attempts to model an optimization trajectory similar to the SIMP method. These approaches have also produced great results that extend to 3D structures. The authors used the initial strain energy and

volume fraction to predict an optimized topology in this work. Additionally, they utilized a separate network to predict that optimized topology's corresponding strain energy, thus training a surrogate model to a Finite Element Analysis algorithm. The predicted topology and corresponding strain energy are then fed back through the network in a cyclic fashion to predict the final density more accurately (Rade et al. 2021). Another approach of data-driven topology optimization methods is the use of generative modeling. While there exist works that attempt to use generative modeling for topology optimization, most of them are still applied to very targeted areas (Rawat and Shen 2019b,a). Hence, there is a need for a general and robust generative design algorithm that is unsupervised and does not use data directly. On the other extreme, data-free machine learning-based topology optimization approaches explore the design space using a neural network by iteratively evaluating the objective function and training the neural network to predict optimal densities (Chandrasekhar and Suresh 2021). However, these approaches do not displace the repeated Finite Element evaluation, thus resulting in high computational cost.

In this work, we explore the generative modeling approach. Specifically, we study the trade-offs associated with using a variant of Generative Adversarial Network (GAN (Goodfellow et al. 2014)) trained strictly using adversarial losses in an unsupervised manner. Generative modeling entails taking some noise from a predetermined random distribution and transforming it into some target (and potentially high-dimensional) distribution. In GANs, the framework comprises two networks (a generator and a discriminator) that compete in a zero-sum game. The generator takes random noise as input and generates some data or images in the context of this work. The generated images are passed to the discriminator and are assigned the probabilities of the generated images being sampled from the target distribution. In essence, the generator tries to fool the discriminator by generating images that resemble the target distribution, while the discriminator tries to differentiate between the generated images and images sampled from the target distribution. For this work, the variant of GAN that we used is the Wasserstein GAN (WGAN) (Arjovsky, Chintala, and Bottou 2017; Gulrajani et al. 2017). A WGAN measures the differences between the true and generated data distributions via the Wasserstein Distance, commonly referred to as the Earth Mover's Distance (EM). Unlike the Jensen-Shannon divergence used in GANs, we can use the EM distance over disjoint distributions since it measures the energy required to turn one distribution into the other.

GANs have been shown to be successful in accomplishing restricted tasks in topology optimization (Oh et al. 2019). These restrictive tasks constrain the problem by either using a dataset composed of very similar structures or slightly editing a given topology such as the iterative design exploration method. GANs have also been used to generate optimal structures given variable physical initial conditions, albeit with a reconstruction loss (Nie et al. 2020). In summary, while GANs have shown great potential for topology optimization, they are still hindered by their relatively uncontrollable adversarial training scheme. This is particularly evident while

training GANs for topology optimization, where the data is heavily constrained by the underlying physical conditions of the structure. This work explores the solution space for given samples' initial physical conditions irrespective of the design constraints. In doing so, we leverage the potential "creativity" of a generator trained strictly off of adversarial losses, opposed to generators explicitly influenced via reconstruction losses.

In summary, our key contributions in this paper are as follows:

1. We propose an unsupervised Wasserstein GAN approach for developing a generative design framework to predict near-optimal structural shapes.
2. We generate a total of 90,000 structures obtained using the traditional SIMP-based topology process.
3. We qualitatively compare our unsupervised Wasserstein GAN approach with the conventional approach in generating shapes in terms of "out-of-distribution" and its ability to satisfy the loads and boundary conditions.

Now, we will cover the details of our method and then show the results.

## Methods

The following section outlines a novel framework for unsupervised generative topology optimization, visualized in Fig. 1, followed by network architectures and loss functions used during training. For comparison, we also trained a vanilla WGAN as a baseline to qualitatively compare our results. Additional background and training details of the baseline framework are in the Appendix.

### WGAN with optimized latent representations

The proposed WGAN with Optimized Latent Representations (WGAN-OLR) framework comprises of two encoder-decoder models and two discriminator models. First, we explain the parts of the network comprising the traditional WGAN and then explain the additional parts added to enhance the performance. The WGAN in this framework is an encoder-decoder model. The encoder of the model takes the initial physical conditions and obtains a latent representation. The noise is drawn from a Gaussian distribution centered at 0 with a variance of 1. The decoder generates a near-optimal density using the noise and the latent representation as input. The entire generator, consisting of the encoder and decoder, is then optimized to generate densities that decrease the Wasserstein distance value output by the discriminator. The standalone WGAN architecture is also shown in the Appendix. In addition to WGAN, WGAN-OLR aims to optimize the latent representations of the initial physical conditions to facilitate a more robust understanding of the potential solution space. For this, we add another encoder-decoder model, shown at the bottom of Fig. 1. This model is trained alongside the WGAN (represented by the encoder$_R$–decoder$_R$ model) in a supervised manner. This supervised encoder-decoder model takes in the initial physical conditions and predicts a density. This model is then optimized via the mean-squared error over the predicted density and the SIMP-optimized density. The encoder component of the generator is optimized to decrease the

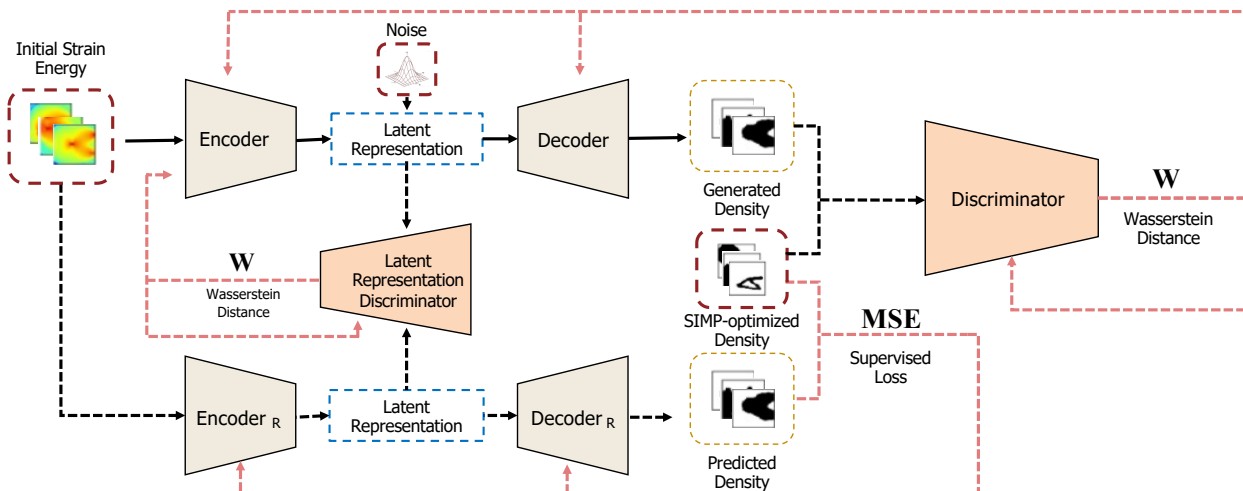

Figure 1: Overall framework for the proposed WGAN with optimized latent representations. Data within maroon boxes denote the input, and the data within the gold box denotes the output. Additionally, black dotted lines denote the forward pass, and salmon-colored dotted lines denote the backward pass during training. During inference, only the components connected by the solid black lines are used to generate new densities.

Wasserstein distance value output by the latent representation discriminator. On the other hand, the density discriminator is optimized to increase the Wasserstein distances between corresponding generated densities and SIMP-optimized densities. Finally, the latent representation discriminator is also optimized to increase the Wasserstein distance between the latent representations produced by the generator's encoder and the supervised model's encoder. The latent representation from the supervised model's encoder is used as the target distribution for the generator's encoder.

To train the proposed WGAN-OLR, we used the RM-Sprop optimizer with a learning rate of 1e-5. We opted for a variable optimization scheme instead of implementing a single update to the generator for a single loss function by utilizing Lagrangian multipliers. The WGAN's generator was updated every third batch, discriminators were updated every single batch, and the supervised model was updated every sixth batch. During experimentation, a variable optimization scheme was shown to be more efficient. We formalize the proposed framework's objective functions below.

$$\min_{G, G_{Enc}} \max_{D, D_{LR}} \mathbb{E}_{x \sim \mathbb{P}_r}[D(x)] - \mathbb{E}_{\hat{x} \sim \mathbb{P}_g}[D(\hat{x})]$$
$$+ \mathbb{E}_{l \sim \mathbb{P}_R}[D_{LR}(l)] - \mathbb{E}_{\hat{l} \sim \mathbb{P}_{lr}}[D_{LR}(\hat{l})] \quad (4)$$

where $D$ is the discriminator, $G$ is the generator and $\mathbb{P}_g$ is the model distribution implicitly defined by $\hat{x} = G(lr, z)$, where $lr$ is the latent representation of the initial physical conditions and $z \sim p(z)$. $G_{Enc}$ is the generator's encoder, $D_{LR}$ is the discriminator for the latent representations and $\mathbb{P}_{lr}$ is the distribution implicitly defined by $\hat{l} = G_{Enc}(SE)$. $\mathbb{P}_R$ is then the probability distribution implicitly defined by $R_{Enc}(SE)$, the latent representation from the supervised model $R$.

$$L_I(R) = \mathbb{E}_{SE,x}[\|x - R(SE)\|_2] \quad (5)$$

The final supervised loss, $L_I(R)$, is defined as the mean squared error between the density predicted, $R(SE)$, and the true SIMP-optimized density $x$. Here the additional supervised encoder-decoder model is defined as $R$.

The encoder and decoder structures used in both frameworks are composed of three levels of down-sampling and three levels of up-sampling, respectively. The density discriminator consists of four down-sampling convolution blocks, each consisting of a 2D convolution operation, followed by 2D instance normalization and leaky ReLU. The discriminator then uses three additional convolution blocks to reduce the number of channels acquired during down-sampling. These convolution blocks are composed of the same operations but with smaller strides to eliminate further down-sampling of the image. The architecture of the Latent Representation Discriminator is composed of two convolution layers and four linear layers, all utilizing a leaky ReLU activation function.

## Data

The training and validation datasets are generated by performing 150 iterations of the SIMP optimization. Each iteration of the SIMP algorithm produces two meshes, one containing that iteration's density and the corresponding mesh representing the compliance. We only require the 2D mesh representing the initial raw compliance and the final optimized density during training. Given that the nodes in each 2D mesh form a regular grid, we can represent those meshes as images, with node values now being pixel values. In total, we generated $\sim 90,000$ structures, each with different randomly generated load values, loading directions, load locations, and a randomly generated set of nodes with fixed displacements. For training, we used 72,000 samples with the remaining $\sim 17,000$ samples used for validation. Each raw compliance image undergoes a log normalization pre-processing operation before being used as input to a network.

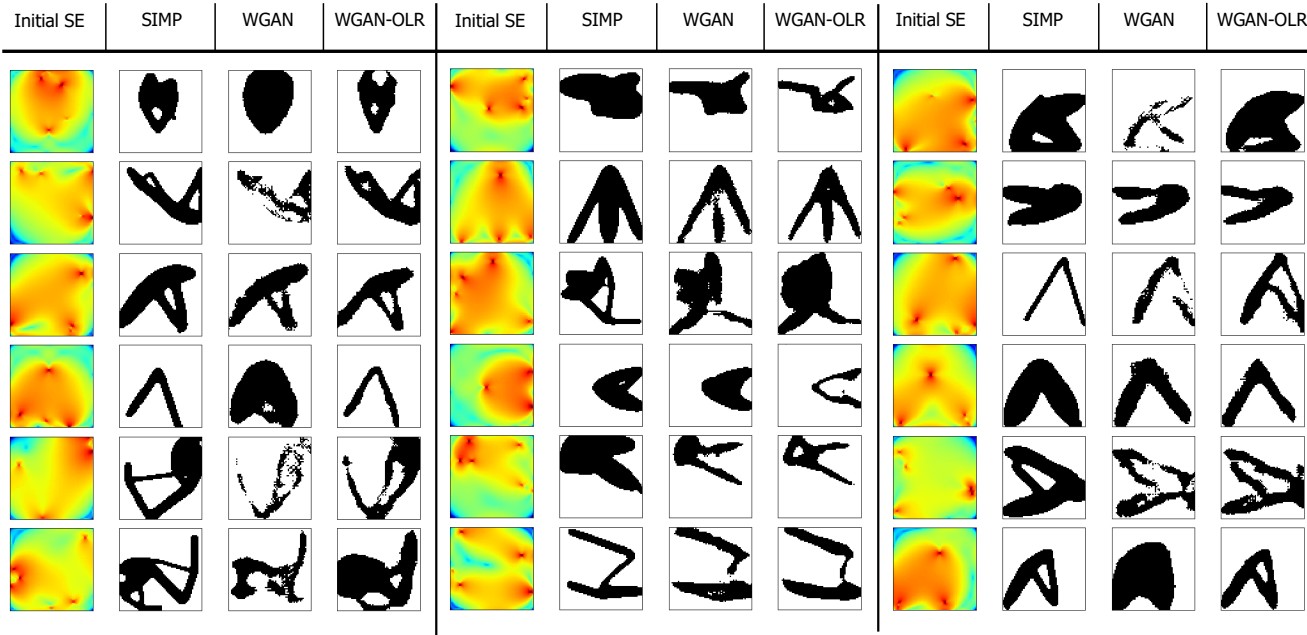

| Initial SE | SIMP | WGAN | WGAN-OLR | Initial SE | SIMP | WGAN | WGAN-OLR | Initial SE | SIMP | WGAN | WGAN-OLR |
|---|---|---|---|---|---|---|---|---|---|---|---|

Figure 2: Sample results arbitrarily divided into three sections. Each section consists of four columns, listed from left to right; Initial strain energy, SIMP-optimized density, baseline WGAN generated density, WGAN-OLR generated density.

## Results and Discussion

We proposed a new generative modeling framework that aims to generate structurally near-optimal densities utilizing the initial physical conditions and without the explicit use of a reconstruction loss. With that being said, evaluating our model off basic distance metrics, i.e., mean absolute and mean squared errors, would not provide meaningful information. Instead, we evaluate our model on the plausibility and uniqueness of designs. In other words, we observe whether the generated densities appears to satisfy the initial physical conditions and, if so, whether it differs from the SIMP-optimized structures. We start by comparing the baseline WGAN generated densities with the SIMP-optimized densities. The WGAN can generate similar densities broadly but typically fails to capture finer regions of each structure. The WGAN either allocates substantially more density or produces disconnected sections at these finer regions of the SIMP-optimized structures. This may be attributed to the learning capacity of the generator itself, or it may allude to the idea that these portions of the solution space are not as integral and may have multiple optimal solutions.

The WGAN-OLR consistently outperforms the baseline WGAN by generating densities that are substantially refined and comparable to the SIMP-optimized densities. This may be attributed to the more robust latent representations the WGAN-OLR's encoder can produce. When comparing the generated densities from both proposed methods, the additional blurriness or noise is not completely unexpected. This may be because both of the proposed methods generate densities in a single shot, whereas the SIMP optimization

algorithm takes 150 iterations to incrementally adjust density placement, providing more opportunities to create smoother structures. Since both proposed methods were trained to learn the distribution of SIMP-optimized structures, there will be an upper bound to the uniqueness of each model's generated densities, which we can also observe from the results.

## Conclusions

In this paper, we introduced completely unsupervised generative models for the generative design/structural topology optimization problem. We proposed a novel method to encode efficient latent representations of given initial physical conditions. This methodology was shown to assist the generator in generating quality alternative solutions to those found via the SIMP optimization algorithm. This work sets a foundation for future works towards more robust structural generative design algorithms. Our future work looks to generate additional data over the same randomly generated initial conditions, i.e., generated load values, loading directions, load locations, and a randomly generated set of nodes with fixed displacements, but utilize different structural topology optimization methods such as level sets or evolutionary algorithms. Doing so would allow the networks to learn a more robust solution space for each set of initial conditions and, in turn, allow the network to generate more unique designs. We also look to incorporate a more quantitative evaluation metric than the qualitative assessment given above. We may accomplish this by incorporating in-the-loop finite methods to evaluate the total compliance of the generated densities.

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

# Appendix

## Baseline WGAN

The baseline framework uses a traditional WGAN with a single generator neural network and a single discriminator neural network. The generator network takes in a tuple consisting of the initial physical conditions in the form of an image and random noise from a Gaussian distribution with a mean of 0 and a variance of 1. The discriminator takes in the generated density, or a known SIMP-optimized density, as input and produces a scalar value as output. The generator is then optimized by the Wasserstein distance, the scalar output by the discriminator. The discriminator is optimized to increase the distance between the Wasserstein distances of a corresponding generated density and SIMP-optimized density. The following equations formalize each objective function.

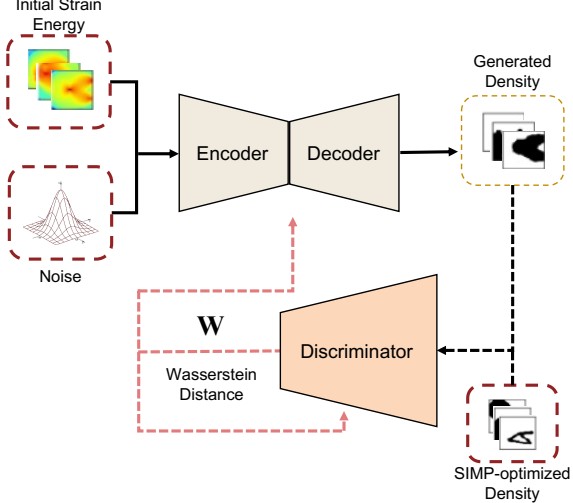

Figure 3: Overall framework for the baseline WGAN.

$$\min_G \max_D \mathbb{E}_{x \sim P_r}[D(x)] - \mathbb{E}_{\hat{x} \sim \mathbb{P}_g}[D(\hat{x})] \qquad (6)$$

where $D$ is the discriminator, $G$ is the generator and $\mathbb{P}_g$ is the model distribution implicitly defined by $\hat{x} = G(SE, z)$, where $SE$ are the initial physical conditions and $z \sim p(z)$.