# OpenReview forum: "Fast Unsupervised Generative Design for Structural Topology Optimization"
_AAAI.org/2022/Workshop/ADAM — AAAI 2022 Workshop ADAM_

### Official Review · Reviewer_9udN · 2021-11-29
**A WGAN architecture for structural TO**

**Rating:** 7
**Confidence:** 4

**Review:**

This paper proposes a new type of architecture for Generative Design, applied in this case to structural topology optimization. The paper introduces readers to back in TO then proposes a dual encoder-decoder WGAN and the training details therein. It demonstrates the method on a SIMP optimized dataset similar to other types of ML-structural TO papers. It provides some visual comparisons of the density fields generated by the proposed architecture along with a simpler WGAN for comparison.

This paper fits the topic of the workshop well and provides some interesting ideas that could lead to useful discussion. I had some reservations about this paper that the authors might address in future efforts:
* The paper refers to itself as a "completely unsupervised" method, but I don't think this is entirely technically accurate, given the supervised losses on the SIMP-Optimized densities as well as the input strain energies (which do encode useful information typically also found in more traditional supervised formulations of this generative design task). This doesn't decrease the merit of the paper overall, but I do think this will confuse readers.
* The paper frequently uses adjectives to describe parts of the method or results but does not quantify this in any meaningful way. For example, in the title "Fast" or in the body "exceptional results" or "substantially outperforms" and many other places, without providing meaningful quantitative measures or a sense of what these adjectives constitute.
* The paper notes in the discussion that the differences between the WGAN and WGAN-OLR may result from "...the learning capacity of the generator itself..." and I will also note that the number of free parameters in the WGAN-OLR network appears to significantly exceed that of the WGAN architecture in Fig. 3. In this way, it is unclear if the changes you see in Fig. 2 are the result of the architectural differences or mearing the significantly increased number of networks and network capacity.

---

### Official Review · Reviewer_XBGk · 2021-11-30
**Review for Fast Unsupervised Generative Design for Structural Topology Optimization**

**Rating:** 7
**Confidence:** 2

**Review:**

The paper mainly proposed a novel Generative Adversarial Network (GAN) framework—Wasserstein GAN with Optimized Latent Representations (WGAN-OLR), which can generate similar structure with target structure that generated from the solid isotropic material with Penalization method (SIMP). This is timely work with novel GAN framework drastically reduces the computational cost of traditional topology optimization. By using deep learning algorithms, authors facilitate the process of coping with data that constrained by underlying physical conditions. Moreover, WGAN-OLR is interesting contribution as it utilized novel encoder and decoder structures that provide more robust latent representations. The WGAN-OLR consistently outperforms the baseline WGAN and generating densities that are comparable to SIMP optimized densities. What could be quantitative metrics to evaluate the robustness of the model (as opposed to visual comparison)? The paper would benefit from the analysis of the computational cost?